# The sGCa Vericiguat Exhibit Cardioprotective and Anti-Sarcopenic Effects through NLRP-3 Pathways: Potential Benefits for Anthracycline-Treated Cancer Patients

**DOI:** 10.3390/cancers16081487

**Published:** 2024-04-12

**Authors:** Vincenzo Quagliariello, Massimiliano Berretta, Irma Bisceglia, Ilaria Giacobbe, Martina Iovine, Vienna Giordano, Raffaele Arianna, Matteo Barbato, Francesca Izzo, Carlo Maurea, Maria Laura Canale, Andrea Paccone, Alessandro Inno, Marino Scherillo, Domenico Gabrielli, Nicola Maurea

**Affiliations:** 1Division of Cardiology, Istituto Nazionale Tumori-IRCCS-Fondazione G. Pascale, 80131 Napoli, Italy; ilaria.giacobbe@istitutotumori.na.it (I.G.); mart.iovine@gmail.com (M.I.); raffaele.arianna@istitutotumori.na.it (R.A.); matteo.barbato@istitutotumori.na.it (M.B.); francesca.izzo@istitutotumori.na.it (F.I.); a.paccone@istitutotumori.na.it (A.P.); n.maurea@istitutotumori.na.it (N.M.); 2Department of Clinical and Experimental Medicine, University of Messina, 98122 Messina, Italy; berrettama@gmail.com; 3Servizi Cardiologici Integrati, Dipartimento Cardio-Toraco-Vascolare, Azienda Ospedaliera San Camillo Forlanini, 00152 Rome, Italy; irmabisceglia@gmail.com; 4ASL NA1, U.O.C. Neurology and Stroke Unit, Ospedale del Mare, 80147 Naples, Italy; carlo.maurea@libero.it; 5U.O.C. Cardiologia, Ospedale Versilia, 55041 Lido di Camaiore, Italy; marialaura.canale@uslnordovest.toscana.it; 6Medical Oncology, IRCCS Ospedale Sacro Cuore Don Calabria, 37024 Negrar di Valpolicella, Italy; alessandro.inno@sacrocuore.it; 7Cardiologia Interventistica e UTIC, A.O. San Pio, Presidio Ospedaliero Gaetano Rummo, 82100 Benevento, Italy; marino.scherillo@libero.it; 8U.O.C. Cardiologia, Dipartimento Cardio-Toraco-Vascolare, Azienda Ospedaliera San Camillo Forlanini, Roma-Fondazione per Il Tuo Cuore-Heart Care Foundation, 00152 Roma, Italy; gabrielli@scamilloforlanini.rm.it

**Keywords:** vericiguat, cGMP, nitric oxide, cancer, doxorubicin, inflammation, cytokines, sarcopenia

## Abstract

**Simple Summary:**

Anthracycline-induced cardiomyopathies and sarcopenia are often seen in cancer patients, affecting their quality of life and overall survival. Translational research aimed to find new cardioprotective strategies is strictly needed in cardioncology. Soluble guanylate cyclase activator vericiguat reduces heart failure rates in patients with reduced ejection fraction. In this study, we highlighted the cardioprotective and anti-inflammatory properties of vericiguat during exposure to anthracyclines and demonstrated its preventive properties of sarcopenia induced by doxorubicin (DOXO), which can be exploited for potential cardioprotective strategies in cancer patients. Furthermore, vericiguat reduces chemokines and cytokines involved in cardiomyopathies through NLR family pyrin domain containing 3 (NLRP-3) pathways in human cardiomyocytes and skeletal muscle cells. The findings that emerged from this study could provide the rationale for further preclinical and clinical investigations aimed at reducing anthracycline cardiotoxicity and sarcopenia in cancer patients.

**Abstract:**

Anthracycline-induced cardiomyopathies and sarcopenia are frequently seen in cancer patients, affecting their overall survival and quality of life; therefore, new cardioprotective and anti-sarcopenic strategies are needed. Vericiguat is a new oral guanylate cyclase activator that reduces heart failure hospitalizations or cardiovascular death. This study highlighted the potential cardioprotective and anti-sarcopenic properties of vericiguat during anthracycline therapy. Human cardiomyocytes and primary skeletal muscle cells were exposed to doxorubicin (DOXO) with or without a pre-treatment with vericiguat. Mitochondrial cell viability, LDH, and Cytochrome C release were performed to study cytoprotective properties. Intracellular Ca^++^ content, TUNEL assay, cGMP, NLRP-3, Myd-88, and cytokine intracellular levels were quantified through colorimetric and selective ELISA methods. Vericiguat exerts significant cytoprotective and anti-apoptotic effects during exposure to doxorubicin. A drastic increase in cGMP expression and reduction in NLRP-3, MyD-88 levels were also seen in Vericiguat-DOXO groups vs. DOXO groups (*p* < 0.001) in both cardiomyocytes and human muscle cells. GCa vericiguat reduces cytokines and chemokines involved in heart failure and sarcopenia. The findings that emerged from this study could provide the rationale for further preclinical and clinical investigations aimed at reducing anthracycline cardiotoxicity and sarcopenia in cancer patients.

## 1. Introduction

Anthracyclines, including doxorubicin, daunorubicin, and epirubicin, are highly effective drugs against cancer, but they can cause damage to the heart and lead to long-term complications [1,2]. Mechanisms of anthracycline cardiotoxicity involve several factors, including the formation of radicals [3], lipid peroxides [4], oxidative stress [5], mitochondrial dysfunction [6], and disruption of the balance of calcium ions within cardiac cells [7]. These factors can ultimately lead to heart muscle damage, resulting in a high risk of cardiomyopathy [8]. Anthracyclines exert cellular damage through increases in NLRP-3, interleukin-1, and interleukin-6 that reduce mitochondrial biogenesis [9,10]. Notably, there is no safe dose of anthracyclines; in fact, it is well described in the literature that doxorubicin exerts a dose-dependent cardiotoxicity; the higher the cumulative dose used in cancer patients, the greater the risk of both acute and chronic cardiomyopathies, as well described in a recent comment of cardiologists and oncologists to the 2022 ESC cardio-oncology guidelines [11].

Cancer patients treated with anthracyclines experienced sarcopenia, a muscle loss process associated with a high risk of heart failure, atherosclerosis, diabetes, and overall mortality [12,13,14,15]. Anthracyclines can cause oxidative stress and inflammation, leading to damage and dysfunction of skeletal muscle tissue [16]: they may also disrupt protein synthesis and promote muscle protein breakdown, resulting in muscle wasting [17]. Cardioprotective strategies are strictly needed in cancer patients to prevent cardiomyopathies induced by anthracyclines [18]. Currently, preventive strategies of doxorubicin cardiotoxicity involve the use of liposomal doxorubicin, dexrazoxane, beta-blockers, sacubitril-valsartan (although it is not recommended in recent cardioncology guidelines for the time being as a protector of cardiotoxicity), ACE inhibitors/ARBs (angiotensin 1-receptor blockers), nutraceuticals and more recently new antidiabetic drugs, called gliflozins [19,20]. However, to date, more translational research is needed to find the best cardioprotective bioactive compound in cardioncology. 

Vericiguat is a soluble guanylate cyclase (sGC) stimulator [21]. It is used for the treatment of heart failure characterized by reduced ejection fraction (called HFrEF) [22]. Vericiguat exerts a cardioprotective effect by targeting the guanylate cyclase enzyme, which plays a role in regulating the synthesis of cyclic guanosine monophosphate (called cGMP), a molecule involved in the relaxation and dilation of blood vessels [23,24]. By stimulating soluble guanylate cyclase, vericiguat increases the production of cGMP, leading to vasodilation and reduced myocardial strain [25]. Moreover, it also improved heart function, decreasing fibrosis and promoting diuresis [26]. In the VICTORIA trial, vericiguat demonstrated its safety in heart failure patients, reducing cardiovascular death or heart failure hospitalization [27,28]. Vericiguat’s vasodilatory effects can help enhance blood flow to skeletal muscle, potentially improving its function and reducing symptoms such as exercise intolerance or sarcopenia [29]. Recent cellular and preclinical studies indicate the beneficial effects of guanylate cyclase on muscle cell survival and prevention of sarcopenia, leading to the potential benefits of pharmacological guanylate cyclase stimulators in muscle loss prevention and treatment [30,31]. 

The present study reported for the first time the preventive effects of vericiguat on skeletal muscle atrophy and cardiotoxicity during exposure to anthracyclines and discussed its potential mechanism of cardioprotection and prevention of sarcopenia in cancer patients. The data of the present study will help advance our understanding of the role of sGCa in the primary prevention of anthracycline-related cardiomyopathies and sarcopenia.

## 2. Materials and Methods

### 2.1. Cell Lines and Pharmacological Treatments

Human cardia cells (called AC-16 cell line) and Primary Skeletal Muscle Cells (HSkMC cells) were purchased from American Type Culture Collection (ATCC^®^, LGC Standards, ATCC, Manassas, VA, USA). Cardiomyocytes were cultured in Gibco^®^ Dulbecco’s modified Eagle’s medium: Nutrient mixture F-12 (DMEM/F12) + 10% *v*/*v* fetal bovine serum (FBS) (HyClone™, GE Healthcare Life Sciences, Chicago, IL, USA) and penicillin associated to Streptomycin at 100 U/mL, Gibco^®^, Gibco, Waltham, MA, USA). Muscle cells were cultured in Mesenchymal Stem Cell Basal Medium associated with Primary Skeletal Cell Muscle Growth Kit (ATCC PCS-950-040 LGC Standards) at a 1:1 ratio, characterized by L-Glutamine (10 mM), rh-EGF (5 ng/mL), rh-FGF-b (5 ng/mL), rh-Insulin (25 µg/mL), Dexamethasone (10 µM), and FBS (4% *v*/*v*). Both cultures were cultured in a proper incubator with 95% air and 5% CO_2_ at 37 °C. For cell viability studies (described in Section 2.2), cells were exposed for 24 h to doxorubicin (DOXO) at 0.5, 1, 10, or 50 µM or pretreated for 1 h with vericiguat at 0.1, 1, or 10 µM, in line with the literature [32]. For the other experiments, both cells were treated with DOXO (1 μM, which is exactly within the therapeutic range, considering that cancer patients treated with DOXO have a systemic anthracycline concentration that ranged from 0.3 to 1 μM [33]) for 24 h with or without a pre-treatment with sGC activator vericiguat for 1 h at 0.1, 1 or 10 μM, in line with the literature [34]. The control group was identified as unexposed cells (culture medium). 

### 2.2. Cell Survival, LDH, and Cytochrome C Release during Exposure to Vericiguat Alone or Combined to Anthracyclines 

In order to examine the impact of vericiguat on the viability of cellular mitochondria, human cardiomyocytes, and muscle cells were plated for 16 h at a density of 150,000 cells per well in 96-well flat-bottom plates. After three washes in PBS, cardiomyocytes and human muscle cells were treated as described in paragraph 2. Survival of the cells: Following treatments, adherent cells were three times washed with PBS at pH 7.4 and then, according to the literature [35], cultured for four hours at 37 °C in 100 μL of an MTS (3-(4,5-dimethylthiazol-2-yl)-5-(3-carboxymethoxyphenyl)-2-(4-sulfophenyl)-2H-tetrazolium) solution (0.5 mg/mL in cell culture medium). Using I-control software (Serial number: 605000008), absorbance measurements were obtained at 450 nm using the Tecan Infinite M200 plate-reader (Tecan Life Sciences Home, Männedorf, Switzerland). The formula used to determine relative cell viability (%) was (A)test/(A)control × 100, where “(A)test” represents the absorbance of the test sample and “(A)control” represents the absorbance of the control cells that were cultured exclusively in culture media. Following the assessment of cell cytotoxicity, we used the Pierce Micro Bicinchoninic Acid (BCA) protein assay kit (Thermo Fisher, Milan, Italy) to determine the total protein concentration [36]. In summary, the cells were prepared as per the manufacturer’s instructions and then incubated for 15 min in 150 μL of the Micro BCA protein assay kit reagent (made in 0.5% *v*/*v* Triton X-100 in PBS) along with an ice-cold PBS wash. Using a plate reader, absorbance at 562 nm was determined. Measurements of cytotoxicity were standardized by the total quantity of protein in each well. Lactate Dehydrogenase release: Using the Cytotoxicity Detection Kit (LDH) (Roche Applied Science, Basel, Switzerland), LDH was measured in the treated cells’ supernatant [37]. A microplate spectrofluorometer was used to measure the signals at 490 nm in order to quantify LDH. Centrifugation was used to gather and harvest cells for cytochrome C extraction. The cell pellet was treated using the Cell Fractionation Kit (Clontech, Palo Alto, CA, USA) in accordance with the manufacturer’s instructions after being washed twice with ice-cold PBS [38]. Douncing the cells 60 times with a tissue grinder and a type A pestle on ice broke them up. The cytosol fraction was obtained by centrifuging the supernatant at 10,000× *g* for 25 min after it had been centrifuged at 700× *g* for 10 min. The Human Cytochrome C ELISA Kit (BioTechne SRL, Milan, Italy) is utilized to quantify cytochrome c in the cytosol fraction of cells. It is a sensitive method of detecting the translocation of cytochrome c from mitochondria into the cytosol (assay range: 0.6–20 ng/mL; sensitivity: 0.31 ng/mL). 

### 2.3. Intracellular Ca^++^ Content

Anthracycline-mediated cardiovascular injuries involve high intracellular calcium levels [39]. Intracellular Ca^++^ in human cardiomyocytes and muscle cells was quantified through the fluorescence dye Fluo-3 AM, according to the manufacturer’s protocol. Both cells were untreated (control) or treated with DOXO for 24 h (1 µM) with and without a pre-treatment for 1 h with sGCa vericiguat (0.1, 1 or 10 µM). Following incubation, cells were loaded with 5 µM Fluo-3 AM and incubated for 30 min at 37 °C in the dark. The excess dye was then removed by washing the cells three times with PBS (pH 7.4). A spectrofluorometer can detect fluorescence when Fluo-3 is chelated with Ca^++^ (ex/em wavelengths of 488 nm and 525 nm, respectively).

### 2.4. Measurement cGMP Intracellular Levels 

Using the Cyclic GMP Complete ELISA Kit (AB133052, AbCam, Milan, Italy), intracellular cGMP levels were measured. In summary, human cardiac and muscle cells were subjected to the 48-h treatment outlined in Section 2.1. Following that, cells were taken out and lysed in lysis buffer, which included a protease inhibitor cocktail, 50 mM Tris-HCl, pH 7.4, 1 mM EDTA, 100 mM NaCl, 20 mM NaF, 3 mM Na_3_VO_4_, and 1 mM PMSF. After centrifuging the lysates at 2500 rpm, the supernatant was used to measure the amount of cGMP in the cells using a selective ELISA kit. Using I-control software (Serial number: 605000008), the measurement was performed at O.D. 405 nm using a Tecan Infinite M200 microplate reader (Tecan Life Sciences Home, Männedorf, Switzerland). Using a five-parameter log-logistic fitted standard curve (using R) created from pure cGMP included in the kit, the cGMP levels in cell lysates were determined [40]. The total protein content was used to normalize the cGMP levels (pmol/mg).

### 2.5. Staining of Intracellular cGMP through CLSM 

Intracellular cGMP levels were stained in human cardiac and muscle cells through a Confocal Laser Scanning Microscope (C1-Nikon). In brief, human cardiac and muscle cells were untreated or treated with DOXO or pretreated with vericiguat for 1 h and subsequently with DOXO for 24 h. Following incubation, the cells were completely cleaned three times with PBS, fixed for twenty minutes with 2.5% glutaraldehyde in PBS, permeabilized for ten minutes with 0.1% Triton-X100 in PBS, and then thoroughly cleaned three times with PBS. The cells were then blocked for 20 min using 1% BSA in PBS. Following the proper washing procedures, the cells were treated for one hour with a 1:100 dilution of purified cGMP Polyclonal Antibody (BS-3892R, Thermo Fisher, Milan, Italy) in 1% BSA. Following PBS washes, cells were incubated with the corresponding secondary antibodies (anti-rabbit) coupled to fluorochromes (FITC) for one hour. The secondary antibodies were diluted 1:1000 in 1% BSA. Concanavalin A Tetramethylrhodamine Conjugate (Invitrogen, Life Technology, Milan, Italy) was used to produce membrane staining, with a final concentration of 100 µg/mL. A Confocal Microscope (C1 Nikon, Tokyo, Japan) fitted with an EZ-C1 Software (Spectral Imaging System C1si Version) for data acquisition and a 60× or 100× oil immersion objective was used to view intracellular cGMP at 488/518 nm and the cell membrane at 555/580 nm [41]. 

### 2.6. Apoptosis Study through TUNEL Assay

Human cardiac and muscle cells were treated, as mentioned in paragraph 2.1, for 24 h. Then they were cultured with CaspaseGlo 3/7 reagent (Promega, Madison, WI, USA) for 30 min at 37 °C in order to examine the anti-apoptotic qualities of vericiguat against doxorubicin cardiotoxicity, as reported in the literature [42]. The activity of caspase-3/7 was then measured with a micro-plate spectrofluorometer.

### 2.7. NLRP-3, MyD-88 Cellular Expression 

For a whole day, the human heart and muscle cells were subjected to the treatment outlined in Section 2.1. Following treatment, cells were taken out and lysed in lysis buffer, which included a protease inhibitor cocktail and 50 mM Tris-HCl, pH 7.4, 1 mM EDTA, 100 mM NaCl, 20 mM NaF, 3 mM Na_3_VO_4_, and 20 mM NaF. Following the guidelines in the literature, lysates were centrifuged at 2500 rpm, supernatants were collected, and NLRP-3 and primary myeloid differentiation response 88 (Myd-88) cellular levels were measured using human NLRP-3 ELISA Kit (OKEH03368, Aviva Systems Biology, San Diego, CA, USA) and human MyD88 ELISA Kit (ab171341, Abcam, Milan, Italy) [43]. In summary, a 96-wellplate (12 × 8 Well Strips) was pre-coated with an antibody against NLRP-3 or MyD88 and blocked. After adding standards or test samples to the wells, they were incubated for one hour. Following 30 s of washing, a biotinylated detector antibody specific to either MyD88 or NLRP3 was applied, incubated, and then cleaned again. After adding and incubating the avidin-peroxidase conjugate, the unbound conjugate was removed by washing. The addition of TMB substrate, which is catalyzed by HRP, resulted in an enzymatic reaction that generated a blue product that became yellow when an acidic stop solution was added. The amount of sample NLRP3 or MYD88 collected in the well was quantitatively correlated with the density of yellow coloring as measured by absorbance at 450 nm. The human NLRP3 ELISA test had a sensitivity of less than 0.078 ng/mL and a detection range of 0.156–10 ng/mL, whereas the human MyD88 ELISA had a sensitivity of less than 10 pg/mL and a detection range of 156.0–10,000 pg/mL.

### 2.8. Cytokines and Growth Factors Assay

Using the ELISA technique, it was possible to measure the intracellular expression of pro-inflammatory cytokines in cardiomyocytes and human muscle cells that are implicated in cardiotoxicity and sarcopenia, such as IL-1β, IL-6, IL-8, CXCL-2, TGF-β, and IL-18 [43]. Following treatments, cells were lysed, and cytokines were measured in accordance with the manufacturer’s indications (Sigma Aldrich, Milan, Italy). This method’s sensitivity was less than 10 (pg/mL), and the test successfully identified cytokines in the 1–32,000 pg/mL range.

### 2.9. Statistical Analysis

Every test involving cells was carried out in triplicate, and the outcomes are displayed as the mean ± standard deviation (SD). Using Sigmaplot software version 12.5 (Systat Software Inc., San Jose, CA, USA), the Student’s *t*-test was used to examine the statistical significance. If there is a significant difference between two data values, the *p*-value is less than 0.05.

## 3. Results

### 3.1. Cytoprotective Properties of Vericiguat

Assessment of mitochondrial dehydrogenase activity through MTS assay clearly evidenced cardiotoxic and skeletal muscle cell damage properties of doxorubicin (Figure 1). Interestingly, the sGCa vericiguat significantly improved cell viability in both human cardiac cells and muscle cells in a concentration-dependent manner (Figure 1A,C). At the higher sGCa concentration (10 µM), cell viability of DOXO-exposed cells (50 µM) was increased by 43.8 and 41.5% for cardiomyocytes and muscle cells, respectively (*p* < 0.001 for both). Similarly, caspase 3/8 activity was drastically increased in both cell lines after doxorubicin therapy (3.4 and 1.89 times for cardiomyocytes and muscle cells, respectively; *p* < 0.001 for both); however, pre-treatment with vericiguat significantly reduced their activity in a concentration-dependent manner (−19.7, −49.5, −60.1% for 0.1, 1 or 10 µM of sGC activator+ DOXO groups vs. only DOXO for cardiomyocytes; (−23.1, −39.2, −52.5% for 0.1, 1 or 10 µM of sGC activator+ DOXO groups vs. only DOXO for human muscle cells). 

### 3.2. Vericiguat Reduces Lactate Dehydrogenases (LDH) and Cytochrome-c Release 

We next aimed to characterize lactate dehydrogenases (LDH) and cytochrome-C release from cardiac cells and human skeletal muscle cells exposed to DOXO with or without vericiguat pre-treatment (Figure 2). Additional evidence for the cytoprotective properties of vericiguat was achieved with the experiment shown in Figure 2. The bar graph in Figure 2A,B shows that DOXO treatment caused a ≈30 times increased release of LDH and cytochrome C in human cardiomyocytes, compared to untreated cells, while, for example, a ≈42.4, 78.2, and 91.8% decreased LDH was detected in cardiomyocytes pre-treated with sGCa at 0.1, 1 and 10 µM, respectively, compared to DOXO groups (*p* < 0.001 for all). Similarly, the bar graph in Figure 2C,D shows that DOXO treatment caused a ≈20–28% increased release of LDH and cytochrome C in human skeletal muscle cells compared to control (untreated cells). For LDH, a significant reduction of ≈22.3, 62.1, and 75.3% was detected in cells pre-treated with sGCa at 0.1, 1, and 10 µM, respectively, compared to the DOXO group (*p* < 0.001).

### 3.3. Vericiguat Reduced Intracellular Ca^++^ Levels during Exposure to Doxorubicin

Intracellular Ca^++^ were significantly increased in cardiac and skeletal muscle cells exposed to DOXO (2166.1 ± 165.3 vs. 267.6 ± 103.3 a.u; for cardiomyocytes; 1665.8 ± 204.3 vs. 462.4 ± 87.7 a.u; *p* < 0.001 for human muscle cells; *p* < 0.001 for all) (Figure 3A); pre-exposure with sCG activator vericiguat at 0.1, 1 and 10 µM, reduced significantly iCa++ levels compared to DOXO groups (relatively to 10 µM sCGa: 944.3 ± 133.1 vs. 2166.1 ± 165.3 a.u (DOXO) for cardiomyocytes and 704.2 ± 134.3 vs. 1665.8 ± 204.3 a.u (DOXO), for human muscle cells, respectively; *p* < 0.001).

### 3.4. Vericiguat Restores cGMP Intracellular Levels during Exposure to Doxorubicin

We hypothesized that the results described in Figure 1 and Figure 2 would be mediated by increasing intracellular cGMP levels in both cell lines. In line with the literature, DOXO therapy drastically reduced cGMP levels in human cells (Figure 3B). Figure 3A shows the ability of cGCa Vericiguat to restore intracellular cGMP levels in human cardiomyocytes (0.43 ± 0.08 pmol/mg of protein for sGCa 10 µM + DOXO vs. 0.11 ± 0.02 pmol/mg for DOXO); the same behavior was seen in human skeletal muscle cells (Figure 3B) (0.62 ± 0.05 pmol/mg of protein for sGCa 10 µM + DOXO vs. 0.27 ± 0.08 pmol/mg for DOXO).

### 3.5. cGMP Levels Staining through Confocal Laser Scanning Microscope Method

Through the Confocal Laser Scanning Microscope, we aimed at verifying the expression and localization of cGMP in cardiomyocytes and human muscle cells pre-exposed or not to vericiguat for 1 h and after to DOXO. Unexposed muscle cells and cardiomyocytes have considerable cGMP levels through the CLSM method (Figure 3C–H) (see green spot); red fluorescence clearly detects plasma membrane. Notably, after exposure to doxorubicin (Figure 3D,G), cGMP expression was reduced in cell cytoplasm of both cells, in line with quantitative data showed in Figure 3B. Instead, pre-exposure to vericiguat significantly increased the fluorescence density of cGMP (Figure 3E,H), indicating its anti-inflammatory properties. These imaging results are in line with the quantitative data reported in Figure 3B. 

### 3.6. Vericiguat Reduces NLRP-3 and MyD88 Levels

To assess whether DOXO can affect the intracellular NLRP-3 and Myd-88 levels, a selective ELISA was performed according to the literature. The results (Figure 4A,C) showed a significantly increased expression of NLRP-3 in DOXO-treated cardiomyocytes and skeletal muscle cells, with an increase of 6.7 and 3.8 times compared to the untreated groups (*p* < 0.001 for both). A similar behavior was seen for intracellular MyD88 levels (Figure 4B,D), with a significant increase of 5.3 and 2.8 times compared to untreated cells (*p* < 0.001 for both). Pre-exposure to sGCa at 0.1, 1 and 10 µM, significantly reduced both NLRP3 and MyD88 levels in cardiomyocytes (−1.3, −3.4 and −5.6 times for 0.1, 1 and 10 µM vs. DOXO groups for NLRP-3 and −1.9, −3.5 and −4.2 times for 0.1, 1 and 10 µM vs. DOXO groups for MyD-88, respectively) and human muscle cells (−1.2, −2.7 and −2.1 times for 0.1, 1 and 10 µM vs. DOXO groups for NLRP-3 and −0.9, −1.4 and −1.6 times for 0.1, 1 and 10 µM vs. DOXO groups for Myd-88, respectively). Taken together, the experiments shown in Figure 4 suggest that, compared to DOXO-treated cells, sGC activator vericiguat exerts strong anti-inflammatory effects through the reduction of two key players of cardiomyopathies and sarcopenia. 

### 3.7. Vericiguat Reduces Intracellular Cytokines and Chemokines Involved in Sarcopenia and Cardiotoxicity

To corroborate the results shown in Figure 4, a quantitative analysis of intracellular IL-1β, IL-6, IL-8, CXCL-2, TGF-β, and IL-18 was performed in both cell lines exposed to DOXO with or without pre-exposure to sGCa. The results (Figure 5) clearly showed a drastic increase in IL-1β, IL-6, IL-8, CXCL-2, TGF-β, and IL-18 in only DOXO-treated cells for both cell lines, compared to control groups (*p* < 0.001 for both). Notably, pre-exposure to sGCa induced significant reductions in concentration-dependent manners. In detail, intracellular IL-1β, IL-6, IL-8, CXCL-2, TGF-β, and IL-18 (pg/mg of protein) from human cardiac cells (Figure 5A–D,F) exposed for 24 h to DOXO with a pre-exposure to sCGa vericiguat, for example, at 10 µM, clearly reduced IL-1β, IL-6, IL-8, CXCL-2, TGF-β, and IL-18 levels compared to DOXO (65.6 ± 8.8 vs. 193.2 ± 12.2 pg/mg of protein for IL-1β in sGCa + DOXO vs. DOXO group; 54.4 ± 11.2 vs. 125.6 ± 5.3 pg/mg of protein for IL-6 in sGCa + DOXO vs. DOXO group; 24.6 ± 13.1 vs. 115.2 ± 13.3 pg/mg of protein for IL-8 in sGCa + DOXO vs. DOXO group; 55.1 ± 13.3 vs. 123.8 ± 17.7 pg/mg of protein for CXCL-2 in sGCa + DOXO vs. DOXO group; 17.8 ± 10.4 vs. 67.7 ± 15.8 pg/mg of protein for TGF-β in sGCa + DOXO vs. DOXO group; 20.3 ± 11.2 vs. 48.8 ± 15.5 pg/mg of protein for 1l-18 in sGCa + DOXO vs. DOXO group; *p* < 0.001 for all). A similar behavior was seen for human skeletal muscle cells (Figure 5 G–I, J-L) with a significant reduction in intracellular IL-1β, IL-6, IL-8, CXCL-2, TGF-β, and IL-18 levels compared to DOXO (62.1 ±22.1 vs. 122.2 ± 20.1 pg/mg of protein for IL-1β in sGCa + DOXO vs. DOXO group; 26.7 ±4.8 vs. 77.8 ± 12.3 pg/mg of protein for IL-6 in sGCa + DOXO vs. DOXO group; 39.7 ±17.3 vs. 80.3 ± 20.3 pg/mg of protein for IL-8 in sGCa + DOXO vs. DOXO group; 38.8 ±18.3 vs. 78.3 ± 21.1 pg/mg of protein for CXCL-2 in sGCa + DOXO vs. DOXO group; 20.7 ±6.8 vs. 55.7 ± 16.6 pg/mg of protein for TGF-β in sGCa + DOXO vs. DOXO group; 15.5 ±5.2 vs. 38.6 ± 16.1 pg/mg of protein for 1l-18 in sGCa + DOXO vs. DOXO group; *p* < 0.001 for all).

## 4. Discussion

The present study demonstrated for the first time that treatment with an sGC activator vericiguat prevented doxorubicin-mediated cardiovascular diseases and sarcopenia in vitro models of cancer-therapeutics-related cardiac dysfunction (CTRD). The anthracycline-based anticancer regimen is still used as therapy for several solid tumors, including breast, sarcoma, and liver adenocarcinomas [44]. Cancer patients treated with anthracyclines are exposed to a high risk of cardiomyopathies, depending on the dose of a drug used and the overall cardiometabolic risk of cancer patients (diabetes, hypertension, obesity, and sarcopenia) [45,46]. Several pathways are involved in anthracycline cardiotoxicity and sarcopenia, including induction of lipid peroxidation, iron-related protein damages, high levels of pro-inflammatory cytokines, and intracellular NLRP-3 (inflammasome) [47,48]. Notably, cancer patients treated with anthracyclines are exposed to detrimental muscle loss (sarcopenia) due to multiple factors, including cancer growth [49], loss of appetite [50], reduced micronutrient absorption [51], increase in sedentary lifestyle [52], a low-protein diet [53] and anticancer drug-induced muscle damages [54]. 

Several drugs and nutraceuticals are currently studied as potential cardioprotective tools in cancer patients [55]. Liposomal doxorubicin regimens, Dexrazoxane hydrochloride, gliflozins (SGLT-2i) [56], and Proprotein convertase subtilisin/kexin type 9 (PCSK9) inhibitors [57] exert significant improvements of cardiac functions in cancer patients treated with anthracyclines but more randomized trials are needed to confirm this point. Great attention is paid to SGLT2i in cancer patients with or without diabetes under anthracycline therapy due to their beneficial effects, including reduction of ferroptosis, reduction of lipid peroxidation, improvement of mitochondrial functions, activation of pAMPK, and reduction of the NLRP3-dependent pathway [58]. 

A very recent study [59] described the involvement of cGMP in doxorubicin-mediated cardiotoxicity, hypothesizing a key role in the reduction of mitochondrial function in the cardiomyocyte and muscle cells. cGMP has several actions, such as a decrease in cardiomyocyte apoptosis, a blunting and/or reversal of cardiac hypertrophy, and protection against ischemia/reperfusion damage. Notably, cardiomyocytes’ cGMP activation reduces cellular calcium, which controls mitochondrial activities. Castro et al. conducted a sophisticated study that examined the possible advantages of cGMP in the cardiomyocyte during ischemia/reperfusion damage, as well as in diabetic cardiomyopathy [60,61]. Moreover, NO-sGC-cGMP signaling activity is decreased in the hearts of HF patients, which results in a decrease in PKG activity, a cGMP target [62]. Riociguat is a sGC stimulant that raises the sensitivity of sGC to NO and stabilizes the binding of NO to sGC [63,64]. Since cGMP signaling is important for several pathophysiological processes, including remodeling and HF, the discovery of novel compounds that target cGMP signaling has resulted in the development of more HF therapies [65]. 

Vericiguat is a novel oral guanylate cyclase activator that is able to improve vascular functions in non-cancer patients with cardiovascular diseases [66]. Randomized clinical trials clearly indicate that vericiguat reduced MACE in patients with heart failure at reduced ejection fraction. Mechanistically, as summarized in Figure 6 vericiguat activates the NO-cGMP pathway in cardiac and skeletal muscle cells, determining several beneficial pathways that counteract pro-inflammatory phenotype induced by anthracyclines. In line with our results, vericiguat showed, for the first time, anti-inflammatory properties in both cell lines, increasing intracellular cGMP levels that reduce NLRP3 and Myd88 intracellular levels (Figure 6). 

These results are in line with the literature; in fact, recent research described the key role of cGMP in the modulation of inflammasome concentration in cancer and myocardial cells [67]. NLRP3 and Myd-88 are key orchestrators of chemotherapy-related cardiovascular diseases (CTRCD) and sarcopenia; cytotoxic properties are principally mediated by high levels of pro-inflammatory cytokines and chemokines, such as IL-1β, IL-6, IL-8, CXCL-2, TGF-β, IL-18 and others [68]. We found that vericiguat is able to change the intracellular cytokine composition in skeletal muscle cells and cardiomyocytes exposed to doxorubicin (Figure 5), inducing an anti-inflammatory phenotype. Myocardial and skeletal muscle cells with high levels of IL-1 and IL-6 are exposed to a greater risk of heart attack and rupture of myofibrils, respectively [69]. Recent clinical work associates high plasma levels of chemokine ligand 2 (CXCL2) with chronic heart failure and sarcopenia [70]. 

Another interesting work found that elevated plasma levels of CXCL2 and IGFBP-1 are negatively associated with lower extremity maximal muscle strength in older patients at hospital admission [71]. Interestingly, a recent review described the involvement of aberrant TGF-β levels in different muscle diseases, including Marfan syndrome, muscular dystrophies, and sarcopenia, confirming the key role of pro-inflammatory chemokines and cytokines in the pathogenesis of skeletal muscle diseases [72]. The same behavior was seen in CTRCD and cardiovascular diseases [73]; in fact, a pro-inflammatory signature of circulating cytokines is associated with a higher risk of cardiovascular mortality and morbidity, MACE, and hospitalization for HF. In line with the literature, anthracyclines induce high levels of intracellular IL-1β, IL-6, IL-8, CXCL-2, TGF-β, and IL-18 both in myocardial cells and skeletal muscle cells. On the contrary, soluble cGC activator verigicuat significantly changed this behavior by imprinting an anti-inflammatory cellular phenotype mediated by cGMP/NLRP-3/Myd88 pathways.

Our study has several limitations: firstly, according to the literature [74], the preventive effect of vericiguat was seen only at a fixed incubation time. However, more studies on the time-dependent effect of sGCa will be necessary. Another limitation of the present study is the absence of preclinical models of anthracycline cardiotoxicity. The primary aim of the present study, albeit in vitro, is to verify the putative beneficial properties of sGCa in cellular models of anthracycline-mediated damages, underlining the possible cGMP-dependent pathways involved. However, subsequent studies on mouse models treated with anthracyclines are strictly necessary in order to evaluate the beneficial effects of verciguat on cardiac functions.

## 5. Conclusions

Anthracycline-mediated cardiotoxicity still remains a CTRCD of clinical relevance in oncology. Sarcopenia is frequently seen in cancer patients, induced both by cancer diseases and chemotherapy, including anthracyclines, affecting cancer patient survival, cardiovascular mortality, and quality of life. Currently, sacubitril-valsartan, SGLT2i, and dexrazoxane are used as preventive and therapeutic strategies for CTRCD in cancer patients, but more large trials are needed to establish the best cardioprotective strategy. To date, Vericiguat, a cGMP activator, has demonstrated significant cardiovascular benefits in patients with HF at reduced ejection fraction. The present study, for the first time, indicates Vericiguat as a new potential cardioprotective strategy against doxorubicin-mediated cardiotoxicity and sarcopenia and highlights the intracellular pathways involved. Vericiguat restored cGMP levels in cardiomyocytes and human skeletal muscle cells, which increased mitochondrial functions and reduced NLRP-3 and Myd-88 expression. These effects induce an anti-inflammatory phenotype in both cardiac and muscle cells during exposure to doxorubicin, indicating potential preventive use of cGMPa in cancer patients aimed to prevent CTRCD and sarcopenia.

## Figures and Tables

**Figure 1 cancers-16-01487-f001:**
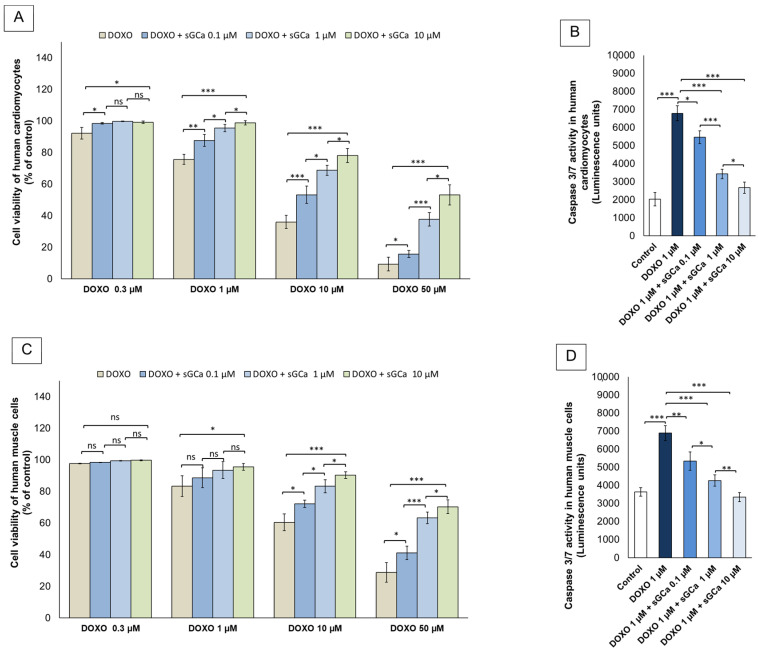
GCa vericiguat reduced cytotoxicity and apoptosis of human cardiac cells and human skeletal muscle cells exposed to anthracyclines. Cell viability and Caspase 3/expression of human cardiomyocytes (AC-16) (**A**,**B**) and primary skeletal muscle cells (HSkMC cells) (**C**,**D**) after 24 h of incubation with doxorubicin (DOXO) alone or pre-treated with sGCa Vericiguat. (**A**,**C**) cell viability of cells treated for 24 h with DOXO (0.5, 1, 10, and 50 µM) or pretreated for 1 h with sGCa Vericiguat (0.1, 1, or 10 µM) and then incubated with DOXO for 24 h. (**B**,**C**) cells untreated (control) or treated with DOXO 1 µM or pretreated for 1 h with sGCa Vericiguat (0.1, 1, or 10 µM) and then incubated with DOXO for 24 h. Error bars depict means ± SD (*n* = 3). Statistical analysis was performed using paired *t*-test *** *p* < 0.001.** *p* < 0.01.* *p* < 0.05, ns: No Significance.

**Figure 2 cancers-16-01487-f002:**
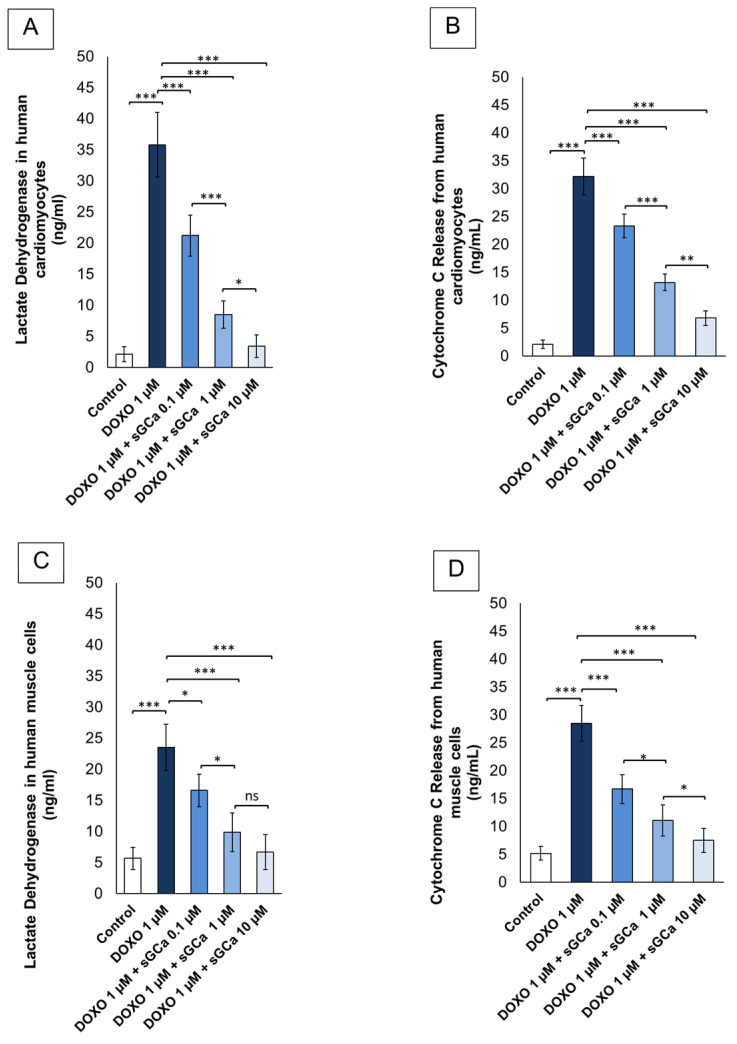
CGa vericiguat exerts cytoprotective properties in human cardiac cells and skeletal muscle cells exposed to subclinical anthracycline levels. Lactate dehydrogenase (**A**,**C**) and cytochrome C release (**B**,**D**) from human cardiac cells (AC 16 cell line) and primary skeletal muscle cells (HSkMC cells) (control) or treated for 24 h with DOXO 1 µM or pretreated for 1 h with sGCa Vericiguat (0.1, 1 or 10 µM) and then incubated with DOXO for 24 h. LDH was quantified in the supernatant of treated cells through a Cytotoxicity Detection Kit (LDH). Signals were quantified using a microplate spectrofluorometer at 490 nm for LDH quantification. For cytochrome c quantification in the cytosol fraction of the cells, a human cytochrome c ELISA Kit, which provides an effective means for detecting cytochrome c translocation from mitochondria into cytosol, was used. Error bars depict means ± SD (*n* = 3). Statistical analysis was performed using paired *t*-test *** *p* < 0.001.** *p* < 0.01.* *p* < 0.05, ns: No Significance.

**Figure 3 cancers-16-01487-f003:**
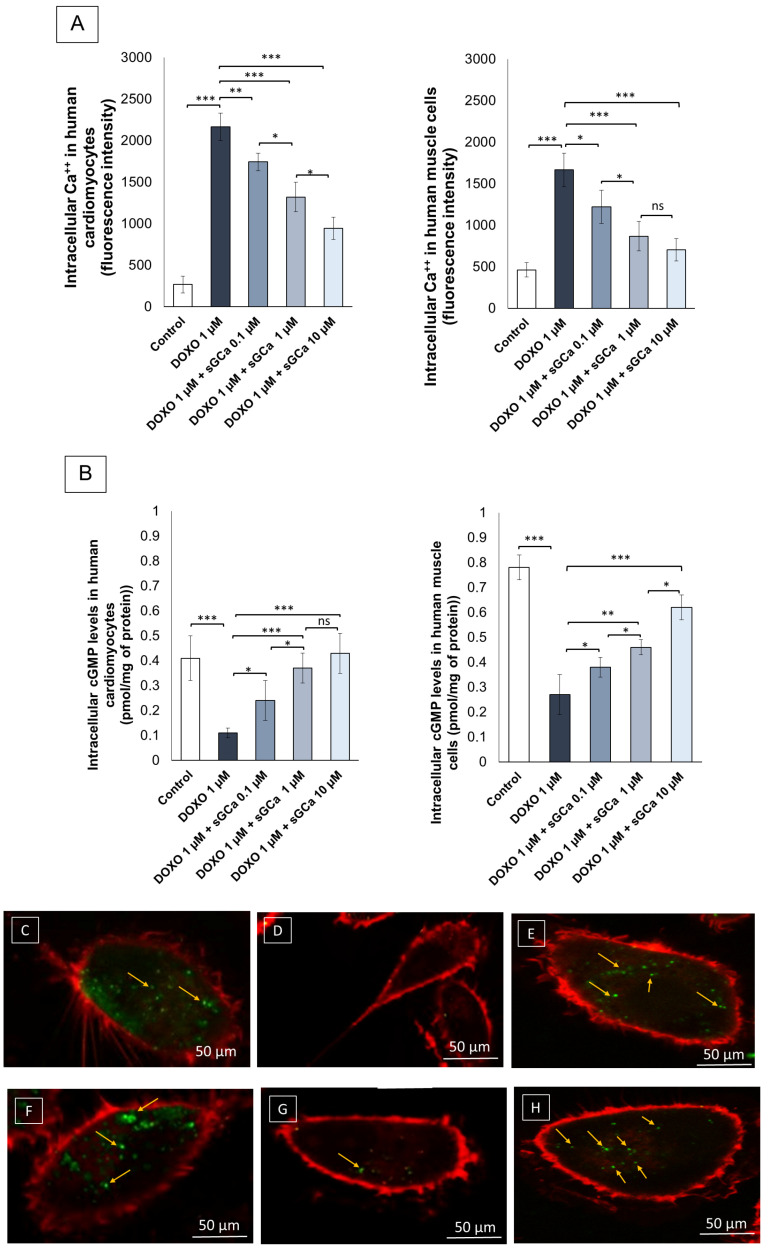
Vericiguat is able to reduce intracellular Ca^++^ and increase cGMP levels in human cardiomyocytes and skeletal muscle cells exposed to subclinical anthracyclines concentrations. (**A**) Intracellular Ca^++^ levels (fluorescence intensity) from human cardiac cells (AC16 cell line) and primary skeletal muscle cells (HSkMC cells) untreated (control) or treated for 24 h with DOXO (0.5, 1, 10, and 50 µM) or pretreated for 1 h with sGCa Vericiguat (0.1, 1 or 10 µM) and then incubated with DOXO for 24 h. (**B**) Intracellular cGMP levels (pmol/mg of protein) from human cardiac cells and muscle cells untreated (control) or treated for 24 h with DOXO (0.5, 1, 10, and 50 µM) or pretreated for 1 h with sGCa Vericiguat (0.1, 1 or 10 µM) and then incubated with DOXO for 24 h. Error bars depict means ± SD (*n* = 3). Statistical analysis was performed using paired *t*-test *** *p* < 0.001.** *p* < 0.01.* *p* < 0.05. ns: No Significance. (**C**–**H**) cGMP intracellular levels in human muscle cells (up) or human cardiomyocytes (down) untreated (**C**,**F**) or treated for 12 h with DOXO (**D**,**G**) or pretreated for 1 h with sGCa Vericiguat (1 µM) and then incubated with DOXO for 12 h (**E**,**H**) cGMP imaging through confocal laser scanning microscope (CLSM). Membrane staining was performed through Concanavalin-Red (red signals); cGMP staining through Anti cGMP primary Abs + secondary antibody conjugated to FITC (green spots). Yellow arrows indicates green spots related to the cGMP levels in cell lines. Scale bar: 50 µm.

**Figure 4 cancers-16-01487-f004:**
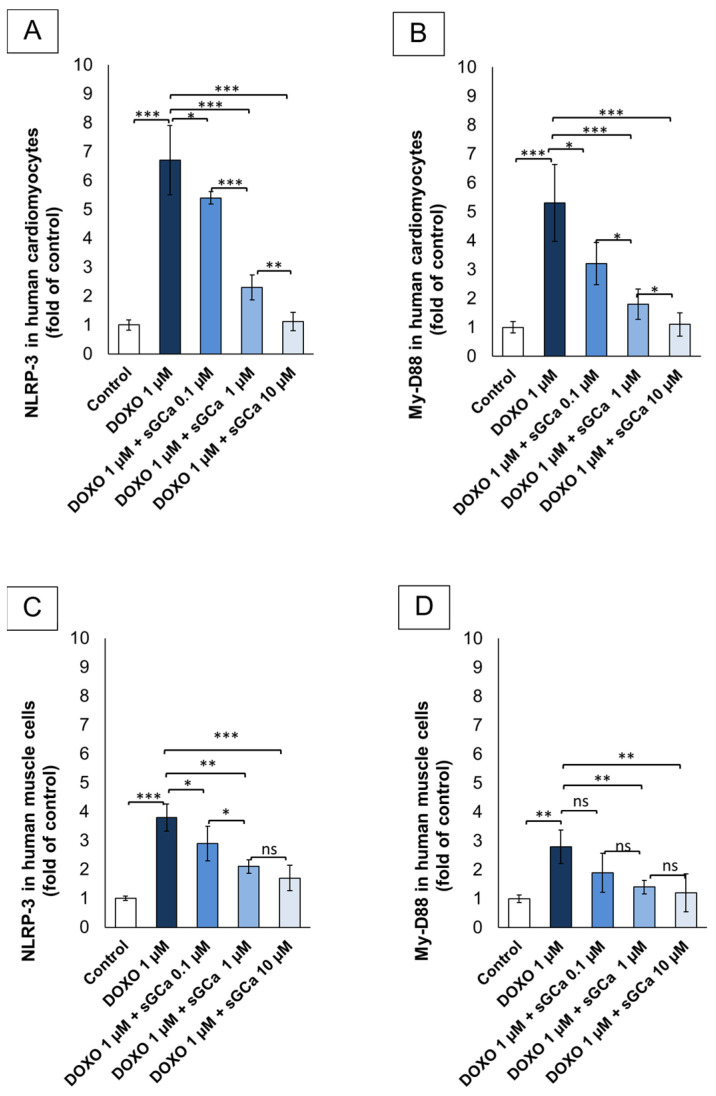
GCa Vericiguat exerts anti-inflammatory properties in human cardiac cells and skeletal muscle cells exposed to subclinical anthracycline levels. Intracellular NLRP-3 levels (fold of control) (**A**,**C**) and MyD-88 (fold of control) (**B**,**D**) human cardiomyocytes (AC-16 cells) (**A**,**B**) and primary skeletal muscle cells (HSkMC cells) (**C**,**D**) untreated (control) or treated for 24 h with DOXO 1 µM or pretreated for 1 h with sGCa Vericiguat (0.1, 1 or 10 µM) and then incubated with DOXO for 24 h. After treatments, cells were harvested and lysed in lysis buffer (50 mM Tris-HCl, pH 7.4, 1 mM EDTA, 100 mM NaCl, 20 mM NaF, 3 mM Na_3_VO_4_, 1 mM PMSF, and protease inhibitor cocktail). Lysates were centrifuged at 2500 rpm, and supernatants were collected and analyzed for NLRP-3 and Myd-88 cellular levels through the human MyD88 and NLRP-3 ELISA Kit. Error bars depict means ± SD (*n* = 3). Statistical analysis was performed using paired *t*-test *** *p* < 0.001.** *p* < 0.01.* *p* < 0.05. ns: No Significance.

**Figure 5 cancers-16-01487-f005:**
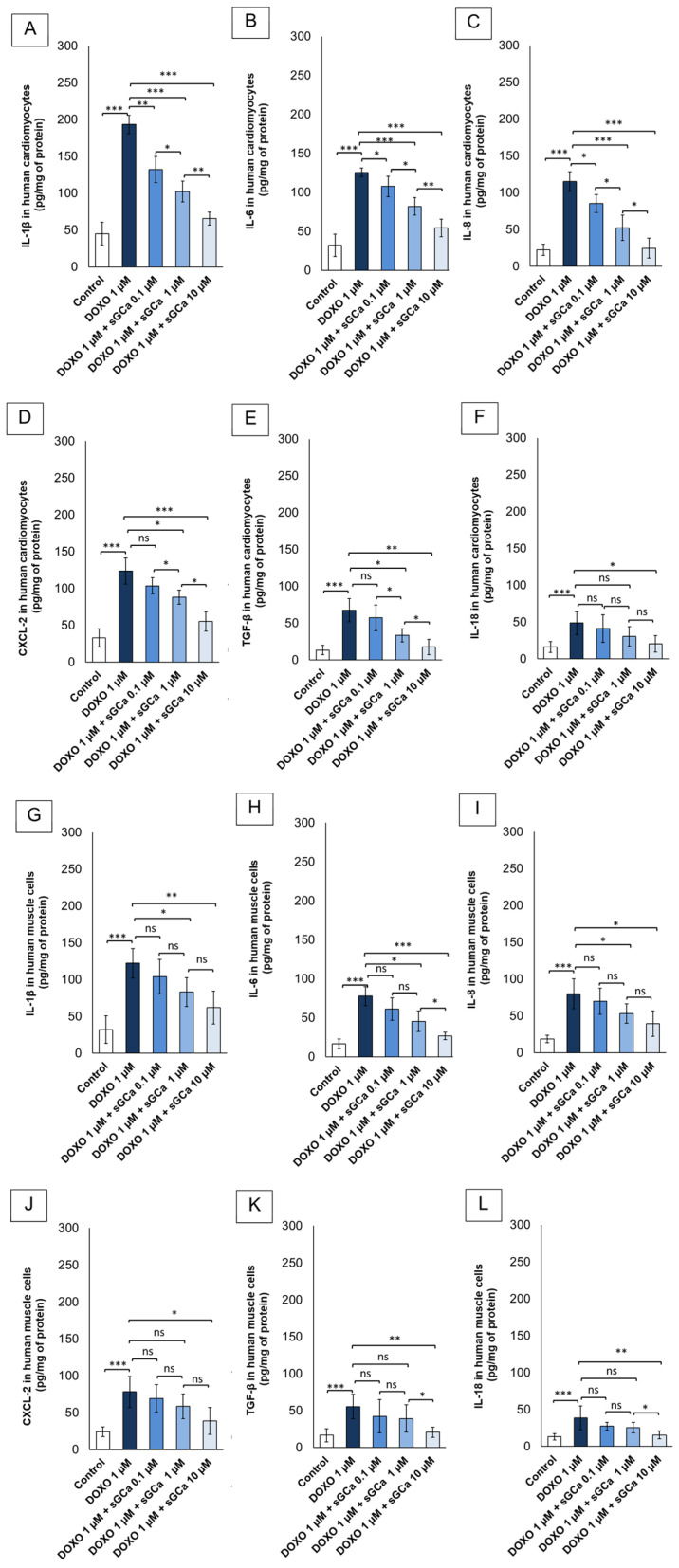
GCa vericiguat is able to reduce pro-inflammatory cytokine and chemokine levels in human cardiac cells and human skeletal muscle cells exposed to subclinical anthracycline levels. Intracellular IL-1β, IL-6, IL-8, CXCL-2, TGF-β and IL-18 (pg/mg of protein) from human cardiomyocytes (AC-16 cells) (**A**–**E**,**F**) and Primary Skeletal Muscle Cells (HSkMC cells) (**G**–**I**,**J**–**L)** untreated (control) or treated for 24 h with DOXO 1 µM or pretreated for 1 h with sGCa Vericiguat (0.1, 1 or 10 µM) and then incubated with DOXO for 24 h. After treatments, cells were lysated, and cytokines were quantified through selective ELISA kits according to the manufacturer’s instructions (Sigma Aldrich, Milan, Italy). Error bars depict means ± SD (*n* = 3). Statistical analysis was performed using paired *t*-test *** *p* < 0.001.** *p* < 0.01.* *p* < 0.05. ns: No Significance.

**Figure 6 cancers-16-01487-f006:**
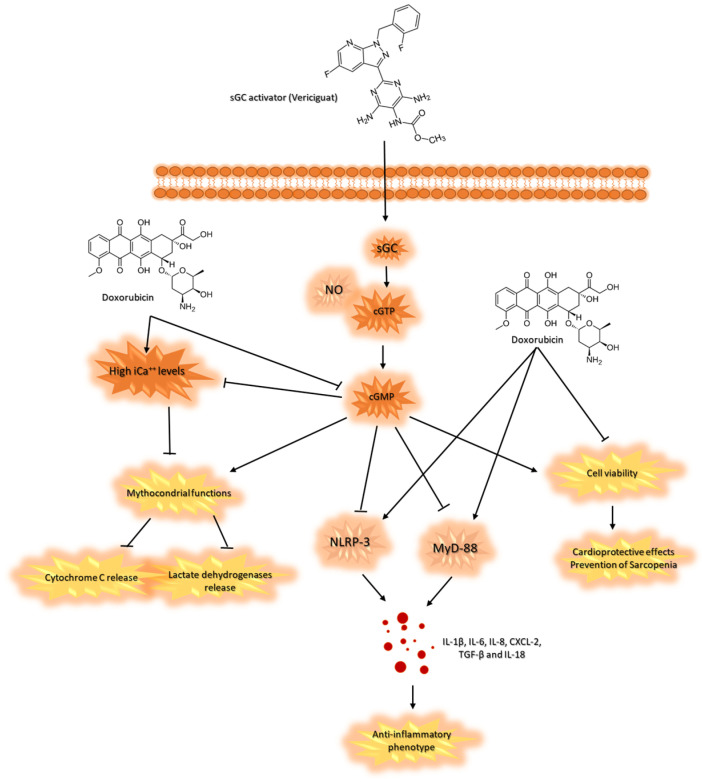
Schematic illustration of sGCa Vericiguat-related cytoprotective properties in human cardiomyocytes and muscle cells exposed to doxorubicin. Vericiguat activates intracellular GC levels, leading to high levels of cGMP in cardiomyocyte and muscle cells. High intracellular cGMP levels improves mitochondrial functions of cardiomyocyte, reducing cytochrome C and lactate dehydrogenase release from cells exposed to doxorubicin. Furthermore, high cGMP levels reduce MyD-88 and NLRP3 expression, leading to reduced IL-1β, IL-6, IL-8, CXCL-2, TGF-β, and IL-18 intracellular levels (strongly enhanced by doxorubicin therapy). Soluble GCa Vericiguat induces an anti-inflammatory phenotype in human muscle cells and cardiomyocytes exposed to doxorubicin, proposing a novel preventive tool for anthracyclines-induced cardiomyopathies and sarcopenia.

## Data Availability

Raw data will be available at https://zenodo.org/records/10603905 (accessed on 1 February 2024).

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
