# Peer review of "The sGCa Vericiguat Exhibit Cardioprotective and Anti-Sarcopenic Effects through NLRP-3 Pathways: Potential Benefits for Anthracycline-Treated Cancer Patients"

_cancers, 2024, doi:10.3390/cancers16081487_

Round 1

Reviewer 1 Report

Comments and Suggestions for Authors

The authors' main objective was to systematically document the in vitro protective effect of vericiguat [Pubchem CID 54674461, soluble guanylate cyclase (sGC) stimulator] against doxorubicin (DOXO, anthracycline, PubChem CID 31703)-mediated cardiotoxicity (in AC-16 cells) and sarcopenia (in HSkMC cells). The authors should consider the following to improve the manuscript´s scientific soundness and uniqueness:

·         General. A) The manuscript´s reading and overall comprehension will improve if it is reviewed by a native English-speaking colleague or by a formal translation agency. B) Include the meaning of each abbreviation (regardless of whether it is common to molecular biologists or not) the first time it is mentioned [e.g. doxorubicin (DOXO) line 37, MTS (3-(4,5-dimethylthiazol-2-yl)-5-(3-carboxymethoxyphenyl)-2-(4-sulfophenyl)-2H-tetrazolium)line 146, angiotensin 1-receptor blockers (ARBs) line 95, bicinchoninic acid (BCA) line 154, primary myeloid differentiation response 88 (MyD88) line 216] and, if possible, reduce its use throughout the manuscript.

·         Title. Quite long. Suggestion: The sGCa Vericiguat exhibit cardioprotective and anti-sarcopenic effects in vitro: Potential benefits for anthracycline-treated cancer patients.   

·         Simple summary. OK. 

·         Abstract. A) Seems quite long. t is advisable to review other articles to check the length and form commonly reported in this journal (e.g. https://doi.org/10.3390/cancers15051567 ). B) It should be more concise without sacrificing important differential results expressed in a more quantitative way (include p-values). C) The cardioprotective and anti-sarcopenia effects/mechanisms should be narrated differentially

·         Introduction. This section is too long and contains unnecessary information. It is suggested to reconstruct in: A) 3-4 paragraphs (i) Epidemiology of muscular/cardiac disorders in cancer derived from the use of anthracyclines, ii) Pharmacology and use of vericiguat (ref 27) and possible muscular effects (smooth, striated), iii ) direct background of the study that highlights the uniqueness of this new study) and, b) in an "effective" way (see: https://www.nature.com/scitable/topicpage/effective-writing-13815989/ ).

·         Methods. OK

·         Results (description). A) This section must be descriptive (succinct statements) of the results found. B) Eliminate introductory and/or argumentative statements supported by bibliography; These could be used in the discussion section.

·         Tables & figures. A) All images and tables should be formatted according to Cancers´ guidelines. B) All figures should be provided at enough size and resolution (>300 dpi) C)) The titles and footnotes must be detailed enough in such a way that they allow an agile and independent reading from the text. C) It is suggested to merge several figures into one to reduce the number of images in accordance with the instructions for authors.

·         Discussion. A) Cancer-therapeutics related cardiac dysfunction (CTRCD)?. B) As recommended in the introduction section, short "effective" paragraphs of 10-15 lines are suggested, C) to the extent possible support your arguments with comparisons to other similar studies.

·         Conclusions. OK

·  References. A) Too many references for an original research article. It is recommended to reduce to no more than 75 references. B) Several references do not have the appropriate format for this journal.

Comments on the Quality of English Language

Moderate changes (grammar & syntax) are neeeded

Author Response

The authors' main objective was to systematically document the in vitro protective effect of vericiguat [Pubchem CID 54674461, soluble guanylate cyclase (sGC) stimulator] against doxorubicin (DOXO, anthracycline, PubChem CID 31703)-mediated cardiotoxicity (in AC-16 cells) and sarcopenia (in HSkMC cells). The authors should consider the following to improve the manuscript´s scientific soundness and uniqueness: General. A) The manuscript´s reading and overall comprehension will improve if it is reviewed by a native English-speaking colleague or by a formal translation agency. B) Include the meaning of each abbreviation (regardless of whether it is common to molecular biologists or not) the first time it is mentioned [e.g. doxorubicin (DOXO) line 37, MTS (3-(4,5-dimethylthiazol-2-yl)-5-(3-carboxymethoxyphenyl)-2-(4-sulfophenyl)-2H-tetrazolium)line 146, angiotensin 1-receptor blockers (ARBs) line 95, bicinchoninic acid (BCA) line 154, primary myeloid differentiation response 88 (MyD88) line 216] and, if possible, reduce its use throughout the manuscript.

  • We thank the reviewer for the useful comments and suggestions aimed to improve the quality of our work. We are agree with you, in line with your suggestions we have modified the manuscript in several parts and improved the readability of the work. Thank you

Title. Quite long. Suggestion: The sGCa Vericiguat exhibit cardioprotective and anti-sarcopenic effects in vitro: Potential benefits for anthracycline-treated cancer patients.   

  • Ok,we have modified the title of the work in line with your suggestion adding only the NLRP3 pathways as possible signalling of GCa –mediated cardioprotective properties, therefore we have changed the title of the work “ The sGCa Vericiguat exhibit cardioprotective and anti-sarcopenic effects through NLRP-3 pathways: Potential benefits for anthracycline-treated cancer patients  (see the underlined parts). Thank you.

Simple summary. OK. 

Abstract. A) Seems quite long. t is advisable to review other articles to check the length and form commonly reported in this journal (e.g. https://doi.org/10.3390/cancers15051567 ). B) It should be more concise without sacrificing important differential results expressed in a more quantitative way (include p-values). C) The cardioprotective and anti-sarcopenia effects/mechanisms should be narrated differentially

  • Ok,we have modified the abstract of the work in line with your suggestions, making it shorter and more readable

Introduction. This section is too long and contains unnecessary information. It is suggested to reconstruct in: A) 3-4 paragraphs (i) Epidemiology of muscular/cardiac disorders in cancer derived from the use of anthracyclines, ii) Pharmacology and use of vericiguat (ref 27) and possible muscular effects (smooth, striated), iii ) direct background of the study that highlights the uniqueness of this new study) and, b) in an "effective" way (see: https://www.nature.com/scitable/topicpage/effective-writing-13815989/ ).

  • Also here, than you. we have modified the introduction of the work in line with your suggestions, through a definition of three areas of description of the key points of introduction. See underlined parts. Thank you.

Methods. OK

Results (description). A) This section must be descriptive (succinct statements) of the results found. B) Eliminate introductory and/or argumentative statements supported by bibliography; These could be used in the discussion section.

  • Sorry for the mistake, we have changed Results and eliminated background information with associated references in this part, making it more homogeneous and readable.

Tables & figures. A) All images and tables should be formatted according to Cancers´ guidelines. B) All figures should be provided at enough size and resolution (>300 dpi) C)) The titles and footnotes must be detailed enough in such a way that they allow an agile and independent reading from the text. C) It is suggested to merge several figures into one to reduce the number of images in accordance with the instructions for authors.

  • Ok, we incorporated three images into one thus reducing the total number of figures from 8 to 6 ( See underlined parts and new fig 3.Thank you

Discussion. A) Cancer-therapeutics related cardiac dysfunction (CTRCD)?. B) As recommended in the introduction section, short "effective" paragraphs of 10-15 lines are suggested, C) to the extent possible support your arguments with comparisons to other similar studies.

  • Yes, we have added the Cancer-therapeutics related cardiac dysfunction (CTRCD) party in discussion. As in introduction, we have modified the discussion, structuring it in a shorter and more readable way. Thank you.

Conclusions. OK

References. A) Too many references for an original research article. It is recommended to reduce to no more than 75 references. B) Several references do not have the appropriate format for this journal.

  • Ok, we have modified the manuscript and eliminated some references. Total number of actual references is 74. Thank you.

Reviewer 2 Report

Comments and Suggestions for Authors

1.       Very good idea for a study- congrats on the idea! There is currently a lack of substances/drugs that could act protectively in muscle damage after anthracyclines- there are only some trials.

The study raises hopes that Vericiguat may act as a protectant in cancer patients. The idea, of course, requires further research.

2.       Currently, preventive strategies of doxorubicin-cardiotoxicity involves the use of liposomal doxorubicin, dexrazoxane, beta-blockers, sacubitril-valsartan, 94 ACE inhibitors/ARBs, nutraceuticals and more recently new antidiabetic drugs

Sacubitril/valasartan is not recommended in recent cathdooncology guidelines for the time being as a protector of cardiotoxicity. Please add that there is no safe dose of anthracyclines. But the higher the dose, the higher the risk of cardiotoxicity. It is necessary to emphasize that the dose of anthracyclines adds up over a lifetime.  Please add the literature:

A practical approach to the 2022 ESC cardio-oncology guidelines: Comments by a team of experts - cardiologists and oncologists.

Leszek P, Klotzka A, BartuÅ› S, Burchardt P, Czarnecka AM, DÅ‚ugosz-Danecka M, Gierlotka M, KoseÅ‚a-Paterczyk H, Krawczyk-Ożóg A, Kubiatowski T, Kurzyna M, Maciejczyk A, Mitkowski P, Prejbisz A, Rutkowski P, Sierko E, SterliÅ„ski M, Szmit S, Szwiec M, Tajstra M, TyciÅ„ska A, Witkowski A, Wojakowski W, Cybulska-Stopa B.Kardiol Pol. 2023;81(10):1047-1063. doi: 10.33963/v.kp.96840. Epub 2023 Sep 3

Author Response

Very good idea for a study- congrats on the idea! There is currently a lack of substances/drugs that could act protectively in muscle damage after anthracyclines- there are only some trials. The study raises hopes that Vericiguat may act as a protectant in cancer patients. The idea, of course, requires further research.

Currently, preventive strategies of doxorubicin-cardiotoxicity involves the use of liposomal doxorubicin, dexrazoxane, beta-blockers, sacubitril-valsartan, 94 ACE inhibitors/ARBs, nutraceuticals and more recently new antidiabetic drugs

Sacubitril/valasartan is not recommended in recent cathdooncology guidelines for the time being as a protector of cardiotoxicity. Please add that there is no safe dose of anthracyclines. But the higher the dose, the higher the risk of cardiotoxicity. It is necessary to emphasize that the dose of anthracyclines adds up over a lifetime.  Please add the literature: A practical approach to the 2022 ESC cardio-oncology guidelines: Comments by a team of experts - cardiologists and oncologists. Leszek P, Klotzka A, BartuÅ› S, Burchardt P, Czarnecka AM, DÅ‚ugosz-Danecka M, Gierlotka M, KoseÅ‚a-Paterczyk H, Krawczyk-Ożóg A, Kubiatowski T, Kurzyna M, Maciejczyk A, Mitkowski P, Prejbisz A, Rutkowski P, Sierko E, SterliÅ„ski M, Szmit S, Szwiec M, Tajstra M, TyciÅ„ska A, Witkowski A, Wojakowski W, Cybulska-Stopa B.Kardiol Pol. 2023;81(10):1047-1063. doi: 10.33963/v.kp.96840. Epub 2023 Sep 3

  • We are very greateful to the reviewer for the compliments and for appreciating the scientific purposes of our manuscript. Thank you very much for the suggestions given, we have made the requested changes (see underlined parts) and we have added the reference that you suggest to the revised manuscript file (see new ref 11). Thank you.